# High-dimensional communication on etchless lithium niobate platform with photonic bound states in the continuum

Zejie Yu [1], Yeyu Tong [1], Hon Ki Tsang [1] & Xiankai Sun [1✉]

Photonic bound states in the continuum (BICs) have been exploited in various systems and found numerous applications. Here, we investigate high-order BICs and apply BICs on an integrated photonic platform to high-dimensional optical communication. A four-channel TM mode (de)multiplexer using different orders of BICs on an etchless lithium niobate ($LiNbO_3$) platform where waveguides are constructed by a low-refractive-index material on a high-refractive-index substrate is demonstrated. Low propagation loss of the TM modes in different orders and phase-matching conditions for efficient excitation of the high-order TM modes are simultaneously achieved. A chip consisting of four-channel mode (de)multiplexers was fabricated and measured with data transmission at 40 Gbps/channel. All the channels have insertion loss <4.0 dB and crosstalk <−9.5 dB in a 70-nm wavelength band. Therefore, the demonstrated mode (de)multiplexing and high-dimensional communication on $LiNbO_3$ platform can meet the increasing demand for high capacity in on-chip optical communication.

[1] Department of Electronic Engineering, The Chinese University of Hong Kong, Shatin, New Territories, Hong Kong. ✉email: xksun@cuhk.edu.hk

**B**ound states in the continuum (BICs) refer to a type of waves that can coexist with continuous waves without any radiation loss[1–9]. This concept was first proposed by von Neumann and Wigner in 1929[1] with a mathematically constructed three-dimensional potential that can support perfectly confined states in a continuous band. Recently, advancement in nanofabrication technologies has triggered fast development of BICs in photonics[10–16], and the demonstrated physical phenomena are being applied to the areas of sensors[17,18], lasers[11], filters[19], and integrated photonics[14,15,20–23]. For integrated photonics, resonances can be found in a single rib waveguide without any cavity structure due to the BIC mechanism[21–23]. In addition, BICs on an integrated photonic platform can be harnessed for realizing low-loss waveguiding by patterning a low-refractive-index material on a high-refractive-index substrate[14,15,20]. Light guided by the low-refractive-index waveguide can be perfectly confined to the region of high-refractive-index substrate under the low-refractive-index waveguide. The BIC-based integrated photonic platform overcomes the challenges in etching of single-crystal materials, and thus enables exploration of their special functionalities on the integrated photonic platform.

Multiplexing and demultiplexing technologies enable transmitting multiple light signals from multiple fiber channels into a single fiber channel and vice versa. The multiple light signals in a single fiber channel will not interfere with each other if different light signals are encoded with different wavelengths, polarizations, or modes. Therefore, (de)multiplexing technologies can significantly enhance data link capacity to meet the increasing demand for huge data transmission in the past decades[24–27]. Wavelength-division multiplexing (WDM) was first developed by making use of light with different wavelengths, which has greatly increased the capacity of data transmission[28]. Mode-division multiplexing (MDM)[26], space-division multiplexing[29], and polarization-division multiplexing[30] were then developed to increase the transmission capacity of a single wavelength, because the available wavelength bands are usually limited by, for example, the transparency window of optical fibers or optical waveguides and the bandwidth of erbium-doped fiber amplifiers. Therefore, optical (de)multiplexers are of fundamental importance for communication with ultrahigh capacity. To date, these multiplexing technologies have been demonstrated mostly on the silicon-on-insulator (SOI) platform[31–34]. However, silicon photonics require doping to form $p–n$ junctions to achieve high-speed modulation on a chip because of the lack of second-order nonlinearity[35]. The thermal effects introduced by $p–n$ junctions restrict the performance and application environment of silicon modulators. $LiNbO_3$ with large electro-optic coefficients is widely used in commercial high-speed modulators with advantages of high linearity[36], high speed[37,38], and high thermal stability[39]. Therefore, realizing (de)multiplexing and high-dimensional communication on the $LiNbO_3$-on-insulator platform can overcome the limitations of the SOI platform.

Here, we have developed a four-channel TM mode (de)multiplexer by using the high-order propagating BICs existing in a low-refractive-index waveguide on a $LiNbO_3$ substrate, demonstrating the application of BICs in high-dimensional optical communication. By harnessing the properties of propagating BICs, the devices were designed and fabricated without etching of $LiNbO_3$, thus simplifying the device fabrication process and reducing the requirement for realizing mode (de)multiplexing in $LiNbO_3$. First, we designed the $TM_0$–$TM_1$, $TM_0$–$TM_2$, and $TM_0$–$TM_3$ multimode directional couplers that can perfectly couple light from the $TM_0$ mode to a high-order mode ($TM_1$, $TM_2$, or $TM_3$) with extinction ratios >12.0 dB in the wavelength range of 1.51–1.58 µm. The fabricated multimode directional couplers demonstrate insertion loss <1.5 dB and extinction ratio >13.0 dB in the

wavelength range of 1.51–1.58 µm. Then, by cascading the multimode directional couplers, we constructed a four-channel TM mode (de)multiplexer, which exhibits measured insertion loss <4.0 dB and crosstalk <−9.5 dB in the wavelength range of 1.51–1.58 µm. Last, we performed high-dimensional data transmission through the fabricated four-channel TM mode (de)multiplexer at 40 Gbps/channel and observed error-free eye diagrams for all the channels.

## Results

**Design of BICs of different orders.** Figure 1a shows the waveguide structure that supports propagating BIC modes with a low-refractive-index material on a high-refractive-index substrate. Figure 1b, c are the modal profiles ($|\mathbf{E}|$) of the TM bound modes of different orders, whose eigenfrequencies lie in the continuous band of the TE modes (see Supplementary Note 1). Usually, the coupling between the TM bound modes and the TE continuous modes (Fig. 1d) introduces power dissipation channels, and thus optical loss, to the TM bound modes. However, the BICs are just exceptions to this conventional wisdom. The coupling between the TE continuous modes and the TM bound modes can be eliminated by destructive interference between the different coupling channels with certain combinations of structural parameters. Figure 1e plots the propagation loss of the $TM_0$, $TM_1$, $TM_2$, and $TM_3$ bound modes as a function of the waveguide width $w$ at the wavelength of 1.55 µm. The propagation loss of the $TM_0$ (solid), $TM_1$ (dashed), $TM_2$ (dotted), and $TM_3$ (dash-dotted) bound modes approaches zero for certain waveguide widths where the desired BICs are obtained. This means that the BICs can be obtained for the TM bound modes in each order just by engineering the waveguide width. In addition, the tolerance of waveguide width for the BICs in each order to maintain negligible propagation loss is as large as several hundreds of nanometers, which can well be accommodated in device fabrication.

Figure 2a shows the proposed four-channel TM mode (de) multiplexer, which is constructed from cascaded multimode directional couplers. $w_0$, $w_1$, $w_2$, and $w_3$ represent the widths of waveguides $wg_0$, $wg_1$, $wg_2$, and $wg_3$, respectively. $g_1$, $g_2$, and $g_3$ represent the gaps between waveguide $wg_0$ and waveguides $wg_1$, $wg_2$, and $wg_3$, respectively. $L_1$, $L_2$, and $L_3$ represent the lengths of coupling from the $TM_0$ mode in waveguide $wg_0$ to the $TM_1$ mode in waveguide $wg_1$, to the $TM_2$ mode in waveguide $wg_2$, and to the $TM_3$ mode in waveguide $wg_3$, respectively. The four-channel TM mode (de)multiplexer has 180° rotation symmetry with respect to its center. Before constructing the TM mode (de)multiplexer, we should design the multimode directional couplers first. The waveguide widths have to be engineered carefully to meet the following two requirements: (1) Low propagation loss. Waveguide $wg_0$ provides low loss for the $TM_0$ mode; waveguide $wg_1$ provides low loss for the $TM_0$ and $TM_1$ modes; waveguide $wg_2$ provides low loss for the $TM_0$, $TM_1$, and $TM_2$ modes; and waveguide $wg_3$ provides low loss for the $TM_0$, $TM_1$, $TM_2$, and $TM_3$ modes. (2) Phase matching between the $TM_0$ mode in waveguide $wg_0$ and the $TM_1$, $TM_2$, and $TM_3$ modes in waveguides $wg_1$, $wg_2$, and $wg_3$, respectively. Figure 2b plots the effective refractive index and propagation loss of the $TM_0$ (blue), $TM_1$ (red), $TM_2$ (black), and $TM_3$ (green) modes as a function of the waveguide width $w$, where the gray areas mark the regions of low propagation loss for the corresponding modes. To satisfy the phase-matching condition, we set the same effective refractive index for the $TM_0$ mode in waveguide $wg_0$, the $TM_1$ mode in waveguide $wg_1$, the $TM_2$ mode in waveguide $wg_2$, and the $TM_3$ mode in waveguide $wg_3$. Therefore, the intersections of the purple solid line and the short-dashed lines in the gray regions in Fig. 2b mark simultaneous satisfaction of the requirements of low

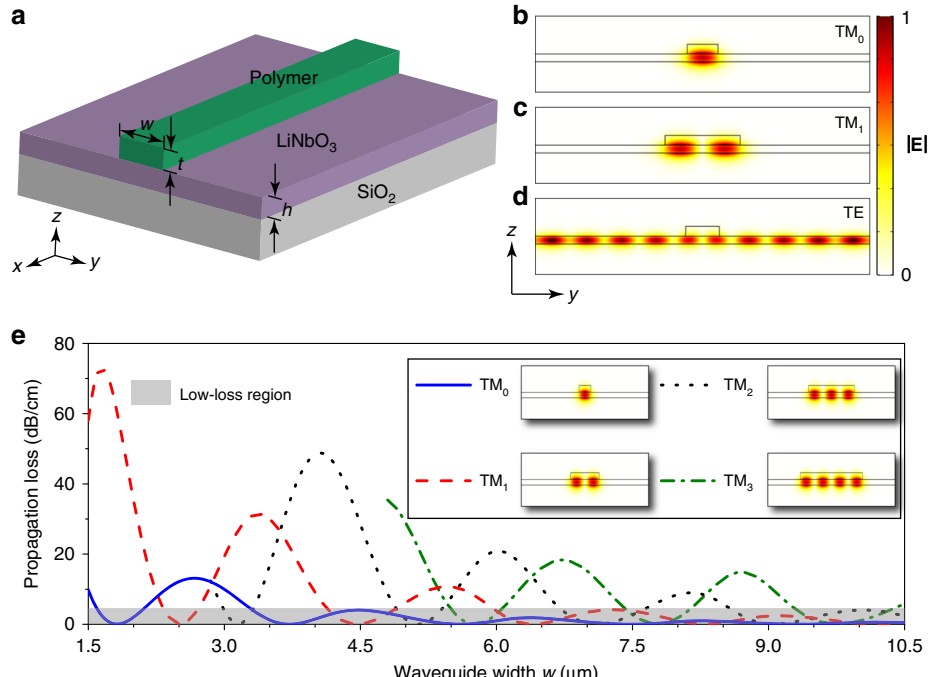

**Fig. 1 Multimode BIC waveguide. a** Waveguide structure composed of a low-refractive-index polymer stripe patterned on high-refractive-index LiNbO$_3$ layer, where $w$, $t$, and $h$ are the waveguide width, the waveguide thickness, and the LiNbO$_3$ layer thickness, respectively. **b–d** Modal profiles $|\mathbf{E}|$ of the TM$_0$ bound mode (**b**), TM$_1$ bound mode (**c**), and TE continuous mode (**d**). **e** Simulated propagation loss of the TM$_0$, TM$_1$, TM$_2$, and TM$_3$ modes in the waveguide as a function of the waveguide width $w$ under the condition of $t = 500$ nm and $h = 400$ nm. For each mode, the BICs exist in the low-loss region as marked in gray.

propagation loss and phase matching, which yields the designed waveguide widths $w_0 = 1.82$, $w_1 = 4.42$, $w_2 = 7.02$, and $w_3 = 9.65$ μm.

Photonic integrated circuits can usually accommodate light within a wavelength band instead of at a single wavelength, so we further simulated the propagation loss spectra for the TM modes in the designed waveguides. Maintaining low propagation loss in a wide wavelength range facilitates applications of the mode (de) multiplexer with WDM technology. Figure 2c–f plot the propagation loss spectra of the TM$_0$ mode in waveguide $wg_0$ with $w_0 = 1.82$ μm, of the TM$_0$ and TM$_1$ modes in waveguide $wg_1$ with $w_1 = 4.42$ μm, of the TM$_0$, TM$_1$, and TM$_2$ modes in waveguide $wg_2$ with $w_2 = 7.02$ μm, and of the TM$_0$, TM$_1$, TM$_2$, and TM$_3$ modes in waveguide $wg_3$ with $w_3 = 9.65$ μm, respectively. The propagation loss of the TM$_0$ mode in waveguide $wg_0$, the TM$_0$ and TM$_1$ modes in waveguide $wg_1$, the TM$_0$, TM$_1$, and TM$_2$ modes in waveguide $wg_2$, and the TM$_0$, TM$_1$, TM$_2$, and TM$_3$ modes in waveguide $wg_3$ can be <0.05, 4.0, 3.5, and 3.0 dB/cm, respectively, at the wavelength of 1.55 μm. The propagation loss of the TM$_0$, TM$_1$, TM$_2$, and TM$_3$ bound modes as a function of both the waveguide width and wavelength is provided in Supplementary Fig. 3.

To obtain the optimal coupling length of the multimode directional couplers, we set $g_1$, $g_2$, and $g_3$ to be 550, 550, and 450 nm, respectively. Figure 3a–c plot the normalized power transmission of the coupled and through ports as a function of the coupling length $L_1$, $L_2$, and $L_3$ at the wavelength of 1.55 μm for the TM$_0$–TM$_1$, TM$_0$–TM$_2$, and TM$_0$–TM$_3$ multimode directional couplers, respectively. It is clear that perfect mode conversion can be achieved with a proper coupling length. Figure 3d–f plot the normalized transmission spectra of the coupled and through ports with the optimal coupling length $L_1 = 125$, $L_2 = 157$, and $L_3 = 143$ μm, respectively. The insertion loss of the TM$_0$–TM$_1$, TM$_0$–TM$_2$, and TM$_0$–TM$_3$ multimode directional couplers can maintain <0.20, 0.25, and 0.45 dB, respectively, in

the wavelength range of 1.51–1.58 μm, with the extinction ratio >17.5, 16.5, and 12.0 dB, respectively. Such a wideband response ensures mode (de)multiplexers constructed from the designed directional couplers to be compatible with WDM, so that they have the potential to achieve WDM and MDM simultaneously. Figure 3g–i plot the simulated electric field profiles ($|\mathbf{E}|^2$) of light coupling from the TM$_0$ mode to the TM$_1$, TM$_2$, and TM$_3$ modes through the TM$_0$–TM$_1$, TM$_0$–TM$_2$, and TM$_0$–TM$_3$ multimode directional couplers, respectively, at the wavelength of 1.55 μm. It is clear that the TM$_0$ mode in the narrower waveguide can be perfectly coupled into the wider waveguides to excite the high-order TM modes.

**Experimental excitation of BICs of different orders.** We fabricated the devices on a 400-nm LiNbO$_3$-on-insulator platform with an underlying silicon handle. The patterned electron-beam resist is transparent in the communication band and thus acts directly as low-refractive-index waveguides on LiNbO$_3$. Figure 4a shows an optical microscope image of the fabricated multimode directional coupler. The close-up is a scanning electron microscope image showing the details of the coupling region. We adopted grating couplers (see Supplementary Note 3) to couple light from an optical fiber into and out of the devices, because the grating couplers are polarization sensitive and thus facilitate high-efficiency excitation of the TM$_0$ mode in the waveguides[40]. The fabricated devices were characterized by spectroscopic measurement of their optical transmission. Light from a tunable semiconductor laser was sent over a single-mode fiber with the polarization state adjusted by a fiber polarization controller, and then coupled into the device under test via the input grating coupler. The transmitted light coupled out of the output grating coupler was collected by a photodetector. Figure 4b–d plot the measured normalized power transmission of the coupled (blue dots) and through (red triangles) ports of the TM$_0$–TM$_1$,

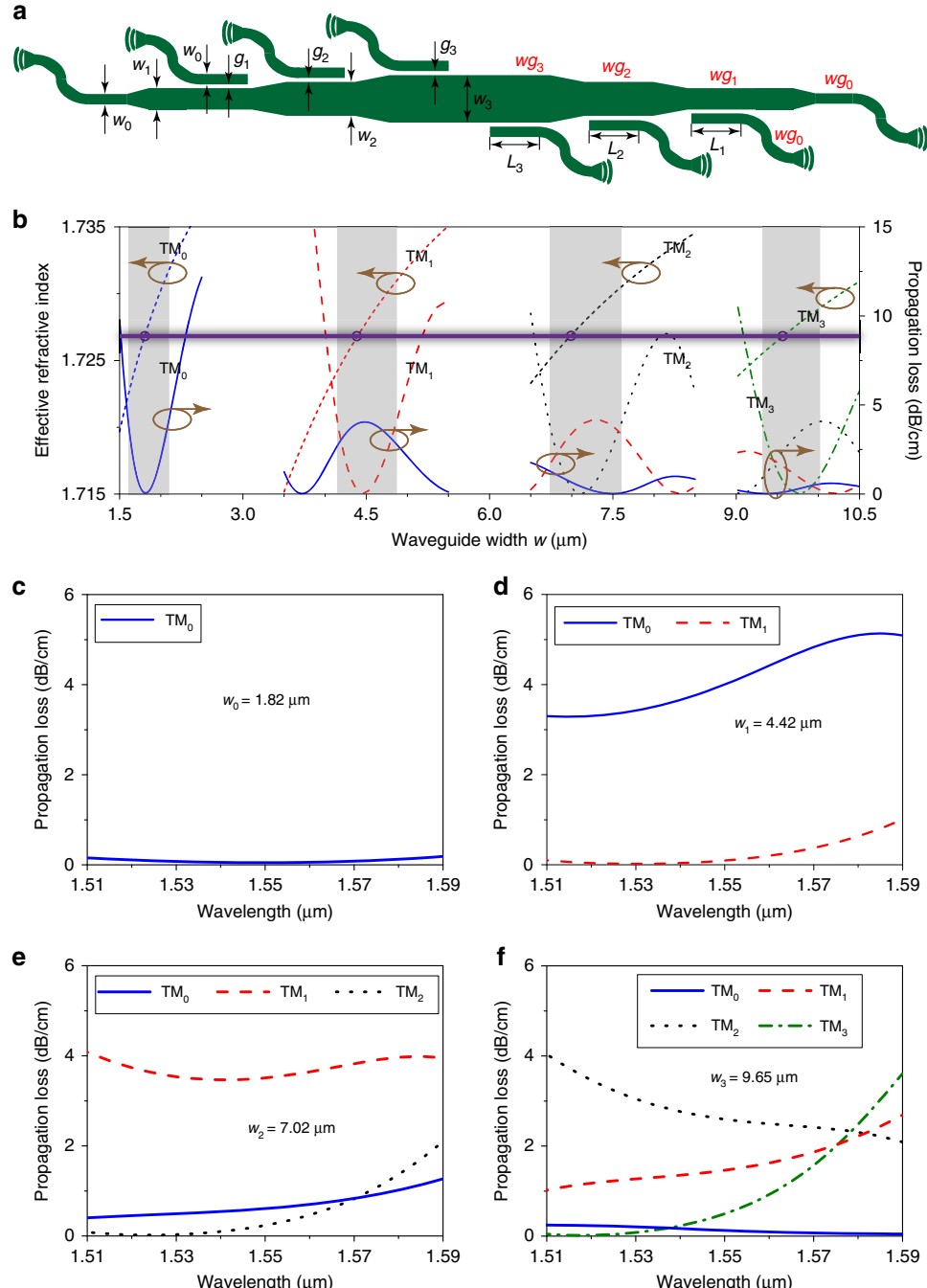

**Fig. 2 Designing mode (de)multiplexer with BICs. a** Illustration of a BIC-based mode (de)multiplexer. $w_0$, $w_1$, $w_2$, and $w_3$ are the widths of waveguides $wg_0$, $wg_1$, $wg_2$, and $wg_3$, respectively. $L_1$, $L_2$, and $L_3$ are the lengths of coupling the $TM_0$ mode from waveguide $wg_0$ to excite the $TM_1$ mode in waveguide $wg_1$, the $TM_2$ mode in waveguide $wg_2$, and the $TM_3$ mode in waveguide $wg_3$, respectively. $g_1$, $g_2$, and $g_3$ are the gaps between waveguide $wg_0$ and waveguides $wg_1$ $wg_2$, and $wg_3$, respectively. **b** Effective refractive index (short-dashed lines) and propagation loss of the $TM_0$ (solid lines), $TM_1$ (dashed lines), $TM_2$ (dotted lines), and $TM_3$ (dash-dotted line) modes as a function of the waveguide width $w$. The four circles on the purple solid line mark the chosen waveguide widths for simultaneous satisfaction of the phase-matching condition and low-loss propagation, which are $w_0 = 1.82$, $w_1 = 4.42$, $w_2 = 7.02$, and $w_3 = 9.65$ μm. **c–f** Propagation loss spectra of the TM BIC modes in waveguides $wg_0$, $wg_1$, $wg_2$, and $wg_3$ with the respective waveguide width $w_0 = 1.82$ μm (**c**), $w_1 = 4.42$ μm (**d**), $w_2 = 7.02$ μm (**e**), and $w_3 = 9.65$ μm (**f**).

$TM_0$–$TM_2$, and $TM_0$–$TM_3$ multimode directional couplers at the wavelength of 1.55 μm. The blue and red solid lines are fits by using a sinusoidal function. For each multimode directional coupler, the maximal power transmission of the coupled port is found to be smaller than 1, which can be attributed to the extra insertion loss due to imperfect device fabrication. Figure 4e–g

plot the measured normalized transmission spectra of the coupled and through ports. The insertion loss of the $TM_0$–$TM_1$, $TM_0$–$TM_2$, and $TM_0$–$TM_3$ multimode directional couplers can maintain <1.2, 0.7, and 1.5 dB, respectively, in the wavelength range of 1.51‒1.58 μm, with the extinction ratio >13.0 dB.

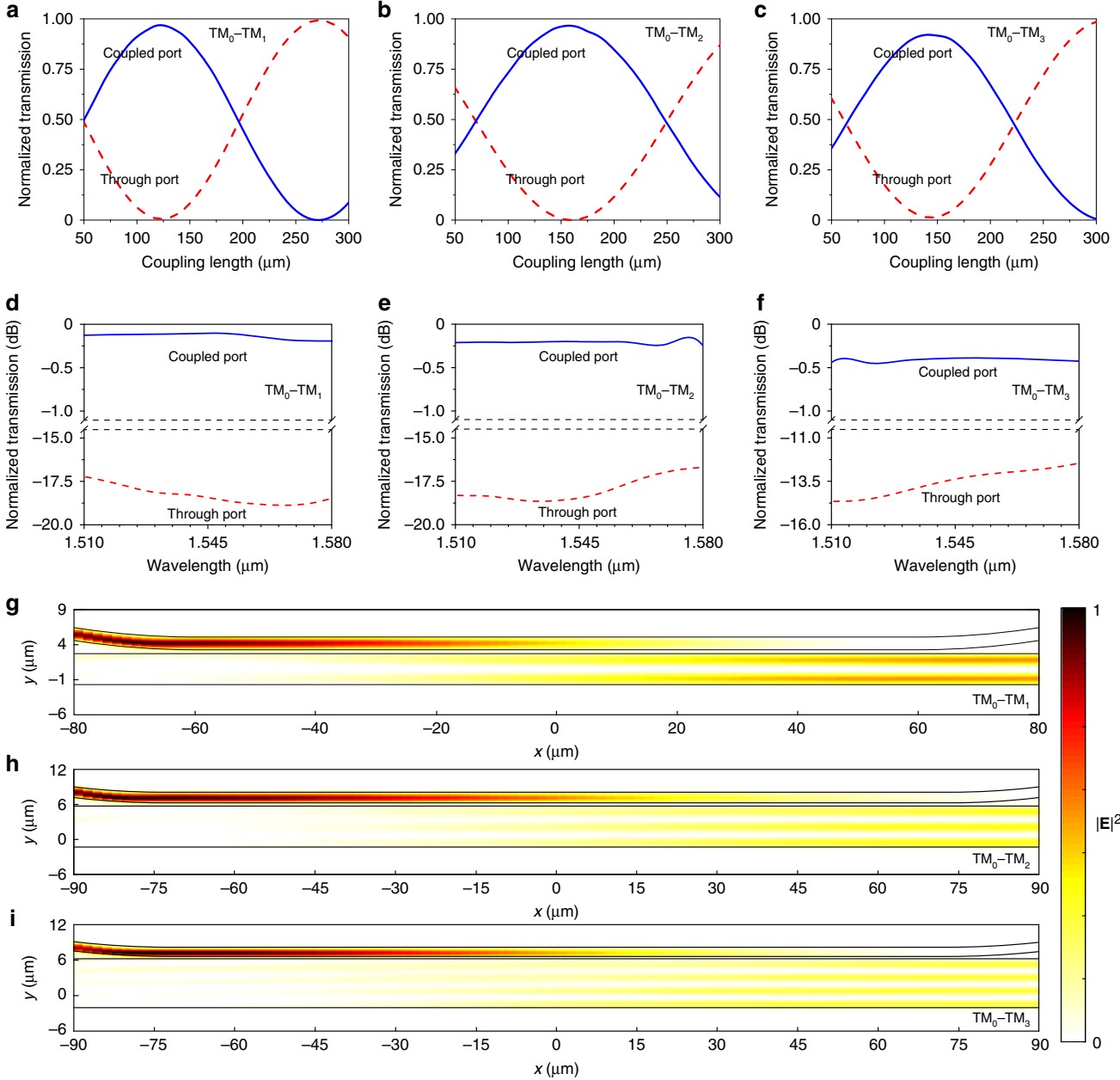

**Fig. 3 Simulated properties of BIC multimode directional couplers. a–c** Normalized transmission of the coupled and through ports at the wavelength of 1.55 μm for the $TM_0$-$TM_1$ (**a**), $TM_0$-$TM_2$ (**b**), and $TM_0$-$TM_3$ (**c**) multimode directional couplers as a function of the coupling length. The $TM_0$ mode can be converted entirely into the $TM_1$, $TM_2$, and $TM_3$ modes through the respective directional couplers. **d–f** Normalized transmission spectra of the coupled and through ports of the $TM_0$–$TM_1$ (**d**), $TM_0$–$TM_2$ (**e**), and $TM_0$–$TM_3$ (**f**) multimode directional couplers with the respective coupling length of 125, 157, and 143 μm. The designed $TM_0$–$TM_1$, $TM_0$–$TM_2$, and $TM_0$–$TM_3$ multimode directional couplers can achieve coupling loss <0.20, 0.25, and 0.45 dB, respectively, with extinction ratio >17.5, 16.5, and 12.0 dB, respectively, in the wavelength range of 1.51–1.58 μm. **g–i** Electric field $|\mathbf{E}|^2$ profiles of light coupled from the $TM_0$ mode to the $TM_1$ (**g**), $TM_2$ (**h**), and $TM_3$ (**i**) modes through the respective multimode directional couplers.

**High-dimensional communication**. We constructed a four-channel TM mode (de)multiplexer by cascading the above demonstrated multimode directional couplers. Figure 5a shows an optical microscope image of the fabricated device. Figure 5b–e show the measured normalized spectra of light transmission from each of the four input ports to each of the four output ports. The insertion loss is 0.8 (<1.7), 2.8 (<3.4), 2.7 (<4.0), and 2.7 (<3.3) dB for the $TM_0$, $TM_1$, $TM_2$, and $TM_3$ modes, respectively, measured at the wavelength of 1.55 μm (in the wavelength range of 1.51–1.58 μm). The crosstalk from other mode channels to the $TM_0$, $TM_1$, $TM_2$, and $TM_3$ mode channels is −14.1, −13.8,

−15.7, and −18.5 dB, respectively, at the wavelength of 1.55 μm and <−9.5 dB in the wavelength range of 1.51–1.58 μm.

To demonstrate the capability of on-chip electro-optic modulation and mode (de)multiplexing, we integrated electro-optic modulators with the mode (de)multiplexer on the same chip as shown in Fig. 6a. Light sent into the four input channels was first modulated by a microcavity electro-optic modulator in each channel, then passed through the mode (de)multiplexer before being directed to the corresponding output channels. Figure 6b shows the experimental setup, where the light from a semiconductor laser at the wavelength of ~1.55 μm was amplified by

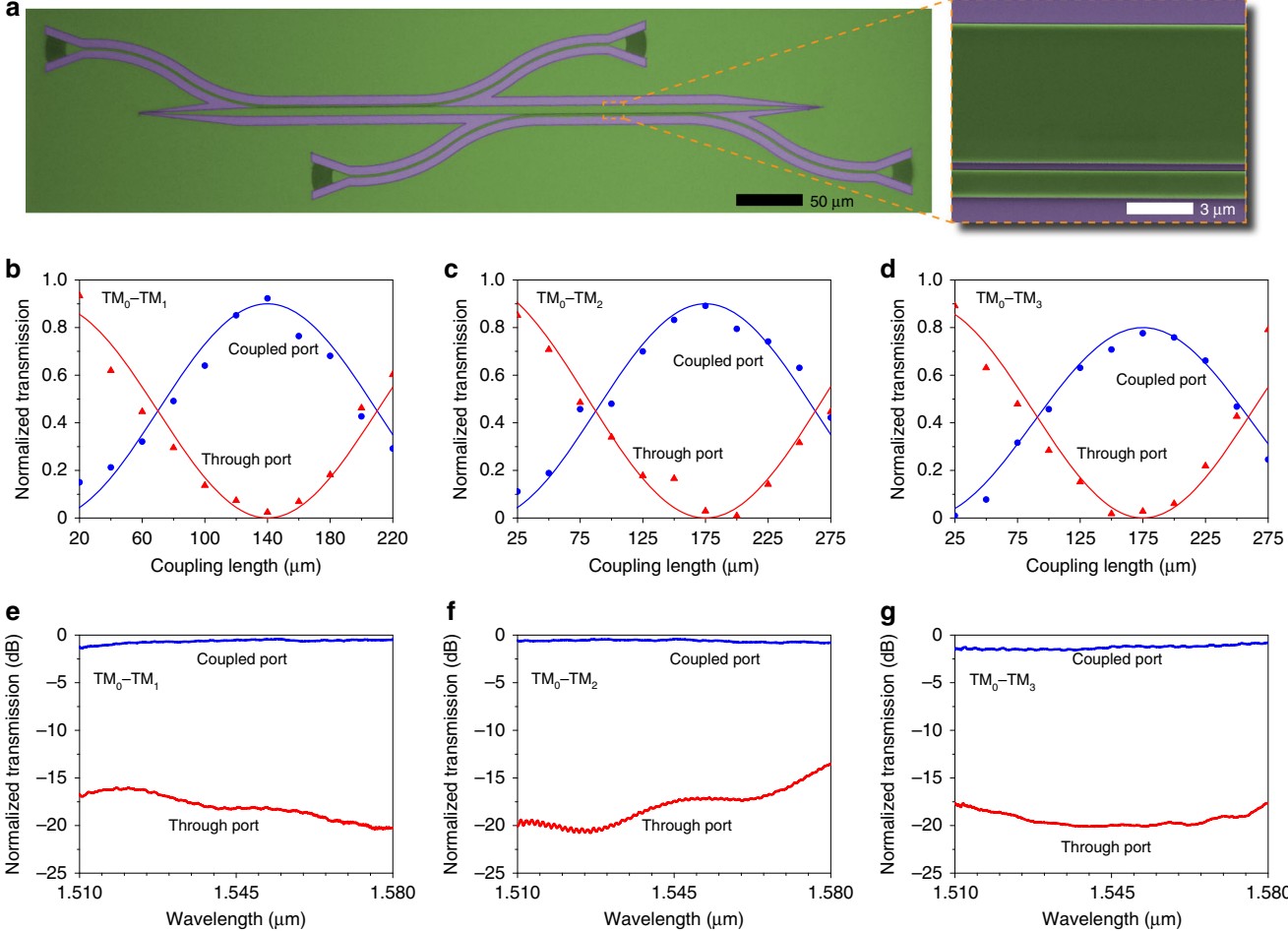

**Fig. 4 Microscopic images and measured properties of BIC multimode directional couplers. a** Optical microscope (left) and scanning electron microscope (right) images of the fabricated multimode directional coupler. The close-up (right) shows the details of the coupling region. **b–d** Normalized transmission at the wavelength of 1.55 μm of the coupled (blue dots) and through (red triangles) ports of the $TM_0$–$TM_1$ (**b**), $TM_0$–$TM_2$ (**c**), and $TM_0$–$TM_3$ (**d**) multimode directional couplers as a function of the coupling length. The solid lines are fits by using a sinusoidal function. **e–g** Normalized transmission spectra of the coupled and through ports of the $TM_0$–$TM_1$ (**e**), $TM_0$–$TM_2$ (**f**), and $TM_0$–$TM_3$ (**g**) multimode directional couplers with the respective coupling length of $L_1 = 140$, $L_2 = 175$, and $L_3 = 175$ μm.

an erbium-doped fiber amplifier and then sent through a fiber polarization controller before being coupled into the device under test. An electrical driving signal from a signal generator was applied to the device under test via a microwave probe. The light transmitted through the device was collected by a high-speed photodetector, which converted the detected optical signal into the electrical domain for monitoring on an oscilloscope. Figure 6c shows the measured signals for the input and output pairs of $TM_0$–$TM_0$, $TM_1$–$TM_1$, $TM_2$–$TM_2$, and $TM_3$–$TM_3$, demonstrating that our fabricated device could achieve both electro-optic modulation and mode (de)multiplexing on a single chip.

Finally, to prove that our fabricated mode (de)multiplexer could support much higher data capacity, we performed high-dimensional data transmission through the fabricated four-channel TM mode (de)multiplexers as shown in Fig. 7a, where the light was modulated outside the chip at a bit rate of 40 Gbps. Figure 7b shows clear eye diagrams observed for the input and output pairs of $TM_0$–$TM_0$, $TM_1$–$TM_1$, $TM_2$–$TM_2$, and $TM_3$–$TM_3$, which clearly indicate that the fabricated mode (de) multiplexer can be used in high-speed optical communication. The crosstalk between different mode channels is discussed in Supplementary Note 2.

## Discussion

In summary, we have investigated the high-order photonic BICs on an etchless LiNbO$_3$ integrated photonic platform, where the high-order TM-polarized bound modes can travel in waveguides constructed by a patterned low-refractive-index material on a high-refractive-index substrate with negligible propagation loss to the TE-polarized continuum. We further harnessed these high-order photonic BICs for high-dimensional optical communication with the aid of a mode (de)multiplexer on the LiNbO$_3$-on-insulator platform. In experiment, the high-order TM modes were excited by the fundamental TM mode through multimode directional couplers. By cascading the multimode directional couplers, we constructed a four-channel mode (de)multiplexer, which has the measured insertion loss of the $TM_0$, $TM_1$, $TM_2$, and $TM_3$ mode channels <1.7, 3.4, 4.0, and 3.3 dB, respectively, and the crosstalk <−9.5 dB in the wavelength range of 1.51–1.58 μm. We also performed high-dimensional data transmission through the four-channel TM mode (de)multiplexer at 40 Gbps/channel and observed error-free eye diagrams for all the channels. The number of (de)multiplexing channels can be further increased, because the BICs for higher-order modes and phase-matching condition can be satisfied simultaneously based on the

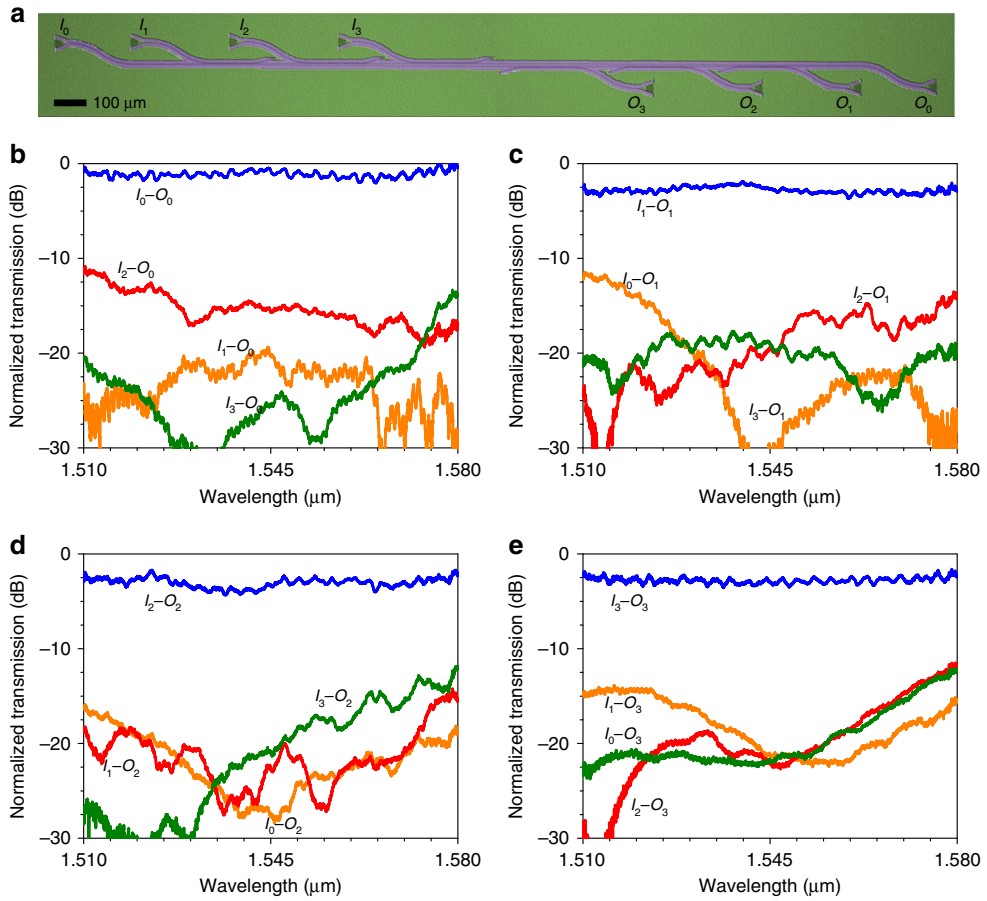

**Fig. 5 Experimental demonstration of mode (de)multiplexing with BICs. a** Optical microscope image of the fabricated mode (de)multiplexer. **b–e** Normalized spectra of light transmission from each of the four input ports $I_0$–$I_3$ to each of the four output ports $O_0$–$O_3$.

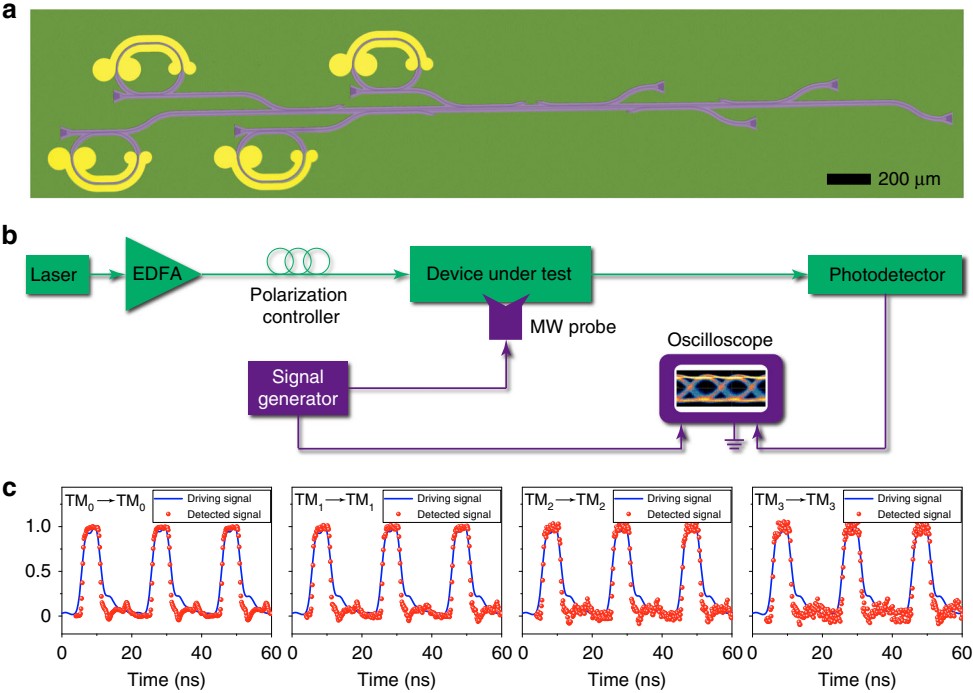

**Fig. 6 Experimental demonstration of on-chip electro-optic modulation and mode (de)multiplexing. a** Optical microscope image of the fabricated mode (de)multiplexer integrated with electro-optic modulators. **b** Experimental setup for measuring electro-optic modulation and mode (de)multiplexing on a single chip. EDFA: erbium-doped fiber amplifier; MW probe: microwave probe. **c** Measured modulated signals in the $TM_0$–$TM_0$, $TM_1$–$TM_1$, $TM_2$–$TM_2$, and $TM_3$–$TM_3$ channels.

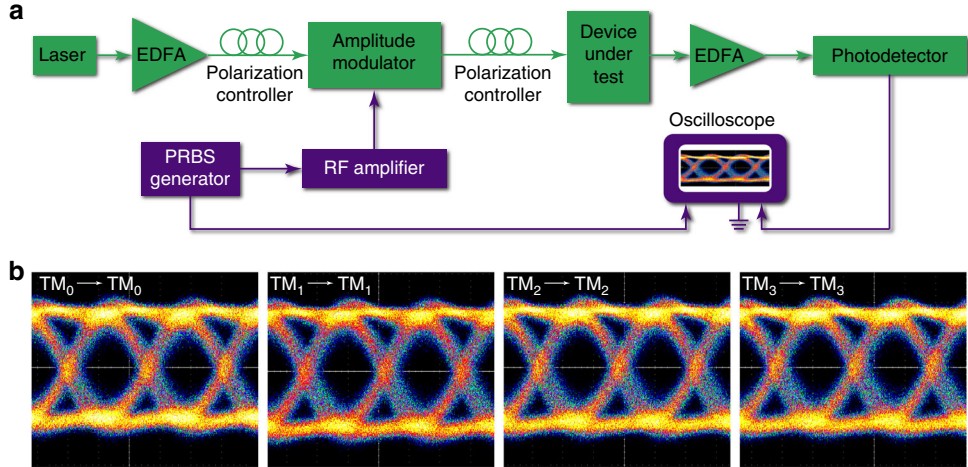

**Fig. 7 Experimental demonstration of high-dimensional data transmission with BICs. a** Experimental setup for measuring data transmission. EDFA: erbium-doped fiber amplifier; PRBS generator: pseudorandom binary sequence generator. **b** Measured eye diagrams of data transmission through a four-channel mode (de)multiplexer at 40 Gbps/channel.

design method in this work. The demonstrated mode (de)multiplexing and high-dimensional communication can significantly enhance the data capacity per wavelength light carrier in a hybrid MDM–WDM optical link on the LiNbO$_3$ platform with the advantages of high speed, high thermal stability, and high linearity.

## Methods

**Simulation**. The effective refractive index and propagation loss were simulated for each mode in the BIC waveguide by using a finite-element method in COMSOL. Eigenmode analysis was adopted in simulating the real and imaginary part of the effective refractive index for each mode. The propagation loss was obtained from the complex effective refractive index.

**Device fabrication**. We fabricated all the devices on a LiNbO$_3$-on-insulator wafer (NANOLN), with a 400-nm-thick LiNbO$_3$ layer on 2-μm-thick silicon oxide. For the passive devices without the function of electro-optic modulation, the fabrication process consists of only one step of electron-beam lithography, which defined the patterns in a 500-nm-thick polymer (electron-beam resist ZEP520A). For the active devices with the function of electro-optic modulation, we first fabricated the electrodes with a lift-off process involving electron-beam lithography and gold deposition, where the thickness of the gold electrodes was 80 nm. Then, we spin-coated a 500-nm-thick layer of polymer (ZEP520A) and performed a second step of electron-beam lithography to pattern the photonic waveguides and grating couplers in the polymer.

## Data availability

The data that support the findings of this study are available from the corresponding author upon reasonable request.

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

## Acknowledgements

This work was supported by the Early Career Scheme (24208915) and the General Research Fund (14208717, 14206318) sponsored by the Research Grants Council of Hong Kong, and by the NSFC/RGC Joint Research Scheme (N_CUHK415/15) sponsored by the Research Grants Council of Hong Kong and the National Natural Science Foundation of China.

## Author contributions

Z.Y. performed the theoretical modeling, device fabrication, device characterization, and data analysis under the supervision of X.S.; Y.T. performed the high-speed communication demonstration under the supervision of H.K.T.; Z.Y. and X.S. wrote the paper with the assistance of all the coauthors.

## Competing interests

The authors declare no competing interests.
