## [Peer Review File · Nature Communications]

Reviewers' Comments:

Reviewer #1:

Remarks to the Author:

The major claim of the manuscript is the use of bound states in the continuum (BICs) as extremely high-Q resonant state in specific application related to multiplexing of photonic channels. It is pleasant to conclude that the aim is achieved. The manuscript demonstrates experimental confirmation of 4-channel TM-mode multiplexer by use of different orders of BICs on lithium niobate platform where waveguides are constructed by a low-refractive-index material on a high-refractive-index substrate.

That certainly introduces a novelty in a few important applications of BICs including lasing, sensing, filtering, giant enhancement of Goos-Hänchen shifts (Feng-Wu et al, Phys. Rev. Applied, 12, 014028 (2019)).

However I am not sure that the paper will influence thinking in the field of light communications by the following reason.

Not all readers are familiar with multiplexing including me. In particular I do not understand how multiplexing can occur in linear systems. I suggest to authors to give a brief introductory and basic principles going beyond listing of references [17-21]. Hopefully that point will be easily improved by authors.

Second comment is the most important. By comparing of the experimental setup in Fig. 1a and Fig. 1a of the paper by E.A. Besus et al, Photonic Res. 6, 1084 (2018) I see that the authors use the same principles for the BIC in the Friedrich-Wintgen scenario for BIC as full destructive interference of two channels with two light polarizations.

Third, the authors cite carelessly papers of BICs. The first experiment by M. Segev group in 2011 followed the first theoretical prediction in photonic crystal (PhC) waveguide coupled with pair of PhC resonators by Bulgakov and Sadreev in PRB 78,075105 (2008). The Fabry-Perot BICs in double array of dielectric rods in paper [6] are still open for experimental realization.

Reviewer comments for paper titled “High-dimensional communication on etchless lithium niobate platform with photonic bound states in the continuum”

September 2019

The manuscript entitled “High-dimensional communication on etchless lithium niobate platform with photonic bound states in the continuum” by Yu *et. al* proposes a new approach for the multiplexing and de-multiplexing (“high-dimensional optical communication”) using an integrated etchless lithium niobate platform that is governed by Bound States in the Continuum (BIC). The authors implement the concept of BIC into their scheme and suggest a structure based on a low refractive index on lithium niobate on insulator (LNOI) platform. This structure creates different potentials for TE and TM modes leading to the existence of TE-mode continuum and TM bound modes. BIC refers to those states (modes) that despite being in the radiation continuum (unlike modes that are confined through total internal reflection (TIR)) they can still be bound without radiation (loss). These states can have infinite quality factors (lifetime). The authors then design a four-port multiplexer by coupling the TM₀ mode to TM₁, TM₂, and TM₃, which is accompanied by experimental results from mode multiplexing (eye diagrams). The general comments about the paper are as follows:

- Authors start their paper with BIC, but they do not profoundly introduce it and make relevant links to their platform. The idea of BIC is exciting but it is not well presented. There are some minor comments such as how TM modes lie within in the continuum of TE modes, but honestly, it seems to be overselling this platform. In spite of the interesting idea, authors fail to motivate the reader towards understanding the underlying physics of this work and their theory. This interpretation of BIC sounds to be in the best case, an alternative explanation of the existing solutions. A profound concept such as BIC requires providing relevant mathematical and physical formulation specific to the platform under study.
- In Figure.1, which is supposed to give the reader an overview of the message of the paper, there is no information about BIC.
- Authors refer to increasing the capacity for increasing demand for huge data transmission (and name the essential techniques) to raise the point that data multiplexing is essential. Then it is noted that these multiplexing techniques have been mostly demonstrated in SOI platform - which is not true (unless they mean for on-chip platforms, which is again not a strong argument). I do not think that there is yet an on-chip solution in commercial use in, for example, any data center. At the end of the paragraph, they motivate the use of LNOI platform in this work by stating its outstanding properties; however, they do not take advantage of this platform at all. The application of LNOI for them is just as a substrate with a larger refractive index than the polymer. Also, (1) proved that LNOI platform has small optical loss down to 2.7 dB/m which is better than the numbers derived from simulations of this work; therefore, the etching issues is also not relevant.
- There exist quite a few papers that implemented the same idea as this paper, such as (2; 3) that they use Silicon Nitride (SiN) on top of LNOI, and they show fast modulations. At the same time, they are other articles such as (4; 5; 6; 7) that use the similar idea of mode coupling (by changing the width of the through-port, gap, and the waveguide (with or without rings). This paper has well combined these two techniques, but the performance is not better than other platforms.

- Authors neglect to provide details on the proposed design (which all can be in supplementary). For example, they forget to mention what gaps they used for the design (g1,g2,g3, and g4). There are also no details on the parameters of the grating couplers and their insertion loss and performance. On the simulation side, I could not find any details on how the propagation loss in figure.1 was calculated. There are also some details of fabrication and measurement missing (E-beam resist, measurement details).
- In the end, for the eye diagram measurement, it is necessary to measure the cross-talk rigorously, writing a number is not sufficient.

According to the comments mentioned above, in my opinion, this manuscript **does not meet** the novelty standards that I would expect for a paper in nature communications.

References

- [1] M. Zhang, C. Wang, R. Cheng, A. Shams-Ansari, and M. Lončar, “Monolithic ultra-high-q lithium niobate microring resonator,” *Optica*, vol. 4, no. 12, pp. 1536–1537, 2017.
- [2] A. N. R. Ahmed, S. Shi, M. Zabolcki, P. Yao, and D. W. Prather, “Tunable hybrid silicon nitride and thin-film lithium niobate electro-optic microresonator,” *Optics letters*, vol. 44, no. 3, pp. 618–621, 2019.
- [3] L. Chang, M. H. Pfeiffer, N. Volet, M. Zervas, J. D. Peters, C. L. Manganelli, E. J. Stanton, Y. Li, T. J. Kippenberg, and J. E. Bowers, “Heterogeneous integration of lithium niobate and silicon nitride waveguides for wafer-scale photonic integrated circuits on silicon,” *Optics letters*, vol. 42, no. 4, pp. 803–806, 2017.
- [4] L.-W. Luo, N. Ophir, C. P. Chen, L. H. Gabrielli, C. B. Poitras, K. Bergmen, and M. Lipson, “Wdm-compatible mode-division multiplexing on a silicon chip,” *Nature communications*, vol. 5, p. 3069, 2014.
- [5] B. Stern, X. Zhu, C. P. Chen, L. D. Tzuang, J. Cardenas, K. Bergman, and M. Lipson, “On-chip mode-division multiplexing switch,” *Optica*, vol. 2, no. 6, pp. 530–535, 2015.
- [6] J. Wang, P. Chen, S. Chen, Y. Shi, and D. Dai, “Improved 8-channel silicon mode demultiplexer with grating polarizers,” *Optics express*, vol. 22, no. 11, pp. 12 799–12 807, 2014.
- [7] Y.-D. Yang, Y. Li, Y.-Z. Huang, and A. W. Poon, “Silicon nitride three-mode division multiplexing and wavelength-division multiplexing using asymmetrical directional couplers and microring resonators,” *Optics express*, vol. 22, no. 18, pp. 22 172–22 183, 2014.

Reviewer #3:

Remarks to the Author:

The manuscript under review is dedicated to the design and experimental demonstration of a 4-channel mode division multiplexer exploiting an unconventional (BIC-based) guiding mechanism. The authors design a multimode polymer waveguide on the lithium niobate platform, in which all the four supported modes are near the BIC condition at different "BIC orders" and thus have low propagation loss and high propagation length. On the basis of this waveguide, the authors design and experimentally demonstrate a 4-channel multiplexer/demultiplexer chip. The fabricated structure has low insertion loss and low channel crosstalk within a relatively broad wavelength range of 70 nm. The fabricated structure, being tested with 40 Gbps/channel data transmission, demonstrates an eye diagram having reasonable quality.

I believe that this paper is, in principle, suitable for publication in Nature Communications since it presents a new important practical application of the BIC effect and, for the first time, demonstrates the possibility of simultaneously using several BICs (or, strictly speaking, several near-BIC modes) of different orders. In addition, the proposed approach allows one to avoid the patterning of the LiNbO₃ layer, which is quite challenging.

However, in my opinion, the manuscript requires a minor revision before it can be accepted for publication. My comments are given below.

1. Bound states in the continuum in ridge structures very similar to the one used as a building block for the proposed (de)multiplexer have been comprehensively studied by several groups. Therefore, in order to place this work in a proper context, I would suggest citing and briefly discussing in the introduction at least the following papers in addition to the cited paper by Zou et al. (Ref. [5] in the manuscript):

- A. P. Hope, T. G. Nguyen, A. Mitchell, and W. Bogaerts. Quantitative analysis of TM lateral leakage in foundry fabricated silicon rib waveguides. *IEEE Photonics Technol. Lett.*, 28(4):493–496, 2016.

- E. A. Bezus, D. A. Bykov, and L. L. Doskolovich. Bound states in the continuum and high-Q resonances supported by a dielectric ridge on a slab waveguide. *Photonics Research*, 6(11):1084–1093, 2018.

- T. G. Nguyen, G. Ren, S. Schoenhardt, M. Knoerzer, A. Boes, and A. Mitchell. Ridge resonance in silicon photonics harnessing bound states in the continuum. *Laser & Photonics Reviews*, (in print):1900035, 2019.

2. Probably, it is worth showing the numerical and experimental results for the transmission of the designed multimode interference couplers (Figs. 3 and 4) on the same plots. It would give the reader a better idea of how close the obtained experimental results are to the results predicted numerically.

3. The authors should comment on the outlook of this work. Can the number of (de)multiplexed channels be increased, or it becomes impossible to maintain the near-BIC condition for a larger number of modes?

4. In the conclusion, the authors state that the demonstrated structure can be used in a "hybrid MDM-PDM-WDM optical link", but do not demonstrate the performance of their structure in the case of TE-polarization. It is clear that quasi-TE modes of the polymer photonic rib waveguide will be confined by the index guiding mechanism, however, a short discussion (probably, in the Supporting Information) would benefit the paper.

Response to Reviewer 1 ---- NCOMMS-19-27126

The major claim of the manuscript is the use of bound states in the continuum (BICs) as extremely high- Q resonant state in specific application related to multiplexing of photonic channels. It is pleasant to conclude that the aim is achieved. The manuscript demonstrates experimental confirmation of 4-channel TM-mode multiplexer by use of different orders of BICs on lithium niobate platform where waveguides are constructed by a low-refractive-index material on a high-refractive-index substrate.

That certainly introduces a novelty in a few important applications of BICs including lasing, sensing, filtering, giant enhancement of Goos-Hansen shifts (Feng-Wu et al, Phys. Rev. Applied, 12, 014028 (2019)).

However I am not sure that the paper will influence thinking in the field of light communications by the following reason.

Response: We thank the reviewer for reading and carefully reviewing our work. We are delighted to see that the reviewer understands and appreciates our work, as shown in his/her comment “That certainly introduces a novelty in a few important applications of BICs...”.

In what follows, we will provide detailed responses to the individual questions.

Comment 1: *Not all readers are familiar with multiplexing including me. In particular I do not understand how multiplexing can occur in linear systems. I suggest to authors to give a brief introductory and basic principles going beyond listing of references [17-21]. Hopefully that point will be easily improved by authors.*

Response: Thanks for the reviewer’s valuable comment. We have briefly introduced what is mode (de)multiplexing in the revised manuscript which is highlighted in red.

Revision details (Main manuscript, Page 2, Paragraph 2, Line 1–5)

Multiplexing and demultiplexing technologies enable transmitting multiple light signals from multiple fiber channels into a single fiber channel and vice versa. The multiple light signals in a single fiber channel will not interfere with each other if different light signals are encoded with different wavelengths, polarizations, or modes. Therefore, (de)multiplexing technologies can significantly enhance data link capacity to meet the increasing demand for huge data transmission in the past decades^[22-25].

Comment 2: *Second comment is the most important. By comparing of the experimental setup in Fig. 1a and Fig. 1a of the paper by E.A. Besus et al, Photonic Res. 6, 1084 (2018) I see that the authors use the same principles for the BIC in the Friedrich-Wintgen scenario for BIC as full destructive interference of two channels with two light polarizations.*

Response: Thanks for the reviewer’s comment. It should be noted that almost all types of demonstrated BICs, including photonic crystal slabs [Nature 499, 188–191, (2013)], waveguide arrays [Phys. Rev. Lett. 107, 183901, (2011)], and rigid waveguides [Photonic Res. 6, 1084 (2018)], are based on destructive interference of different dissipation channels. On top of this principle, we emphasize that our BIC is totally different from that in [Photonic

Res. 6, 1084 (2018)]. First, the BIC in [Photonic Res. 6, 1084 (2018)] is hosted essentially by a rigid waveguide cavity, and the BIC mode resonates in the waveguide rather than being guided by the waveguide. In our work, the BIC modes are guided modes which can be routed arbitrarily by the waveguide. In view of this difference, our BICs can have numerous on-chip applications that cannot be realized by the BIC in [Photonic Res. 6, 1084 (2018)]. For example, our BIC mode can work in devices based on traveling waves, such as power splitters, Mach–Zehnder interferometers, mode (de)multiplexers, etc. Besides these guided-mode applications, our BIC mode can also circulate in photonic microcavities and achieve cavity resonances on a chip.

Comment 3: *Third, the authors cite carelessly papers of BICs. The first experiment by M. Segev group in 2011 followed the first theoretical prediction in photonic crystal (PhC) waveguide coupled with pair of PhC resonators by Bulgakov and Sadreev in PRB 78,075105 (2008). The Fabry-Perot BICs in double array of dielectric rods in paper [6] are still open for experimental realization.*

Response: Thanks for the reviewer’s careful checking. We have removed citations of theoretical work at that sentence and made revisions in the revised manuscript which is highlighted in red.

Revision details (Main manuscript, Page 2, Paragraph 1, Line 5)

...in photonics^[9-14],

Response to Reviewer 2 ---- NCOMMS-19-27126

The manuscript entitled “High-dimensional communication on etchless lithium niobate platform with photonic bound states in the continuum” by Yu et. al proposes a new approach for the multiplexing and de-multiplexing (“high-dimensional optical communication”) using an integrated etchless lithium niobate platform that is governed by Bound States in the Continuum (BIC). The authors implement the concept of BIC into their scheme and suggest a structure based on a low refractive index on lithium niobate on insulator (LNOI) platform. This structure creates different potentials for TE and TM modes leading to the existence of TE-mode continuum and TM bound modes. BIC refers to those states (modes) that despite being in the radiation continuum (unlike modes that are confined through total internal reflection (TIR)) they can still be bound without radiation (loss). These states can have infinite quality factors (lifetime). The authors then design a four-port multiplexer by coupling the TM0 mode to TM1, TM2, and TM3, which is accompanied by experimental results from mode multiplexing (eye diagrams). The general comments about the paper are as follows: ...

According to the comments mentioned above, in my opinion, this manuscript does not meet the novelty standards that I would expect for a paper in nature communications.

Response: We thank the reviewer for reading and carefully reviewing our work. We appreciate that the reviewer recognizes our work as “a new approach for the multiplexing and de-multiplexing...”. However, we respectfully disagree that our work does not meet the novelty standards of *Nature Communications*. In what follows, we will provide detailed responses to the individual comments, hoping that the reviewer can reevaluate our work and appreciate the novelty.

Comment 1: *Authors start their paper with BIC, but they do not profoundly introduce it and make relevant links to their platform. The idea of BIC is exciting but it is not well presented. There are some minor comments such as how TM modes lie within in the continuum of TE modes, but honestly, it seems to be overselling this platform. In spite of the interesting idea, authors fail to motivate the reader towards understanding the underlying physics of this work and their theory. This interpretation of BIC sounds to be in the best case, an alternative explanation of the existing solutions. A profound concept such as BIC requires providing relevant mathematical and physical formulation specific to the platform under study.*

Response: Thanks for the reviewer’s valuable comments. The background information of the BIC-based platform can be found in [Optica Vol. 6, Issue 10, pp. 1342–1348 (2019)] and [Laser Photon. Rev. Vol. 9, Issue 1, pp. 114–119 (2015)]. In our previous work [Optica Vol. 6, Issue 10, pp. 1342–1348 (2019)], we have presented in details what is BIC in photonic integrated circuits (PICs), why we need the BICs in PICs, and how to obtain BICs in PICs. Here, we provide a concentrated version for the reviewer to quickly understand the main idea of BICs in PICs.

The motivation of investigating BICs in PICs. Functional single-crystal materials like LiNbO₃, barium titanate (BTO), and yttrium iron garnet (YIG) have not been effectively utilized in integrated photonics, because these materials either cannot be etched or are difficult to achieve high quality in etched microstructures. One way to overcome this difficulty is fabricating the PICs by using the single crystal as a substrate and another

fabrication-friendly material with a lower refractive index for patterning microstructures. In conventional wisdom, this approach is considered infeasible because the high-refractive-index single-crystal material does not allow efficient confinement of photons in the low-refractive-index material atop. However, by adopting BICs in the integration scheme we can solve this fundamental problem, which has been experimentally demonstrated in our previous work [Optica Vol. 6, Issue 10, pp. 1342–1348 (2019)]. We surprisingly find that light can be efficiently confined by a low-refractive-index waveguide on a high-refractive-index substrate through engineering the structure of the low-refractive-index waveguide. However, in our previous work, all the BIC modes are fundamental modes. In this work, we make exploration to the next step and investigate high-order BIC modes, because the high-order modes are of crucial importance for on-chip applications with increased data capacity. In addition, we harness these high-order BIC modes and demonstrate the application of mode (de)multiplexing.

An intuitive way of understanding and obtaining the BICs in PICs. Defying the conventional wisdom, the BIC mechanism predicts zero propagation loss in a waveguide structure as shown in Fig. R1 where a low-refractive-index waveguide (purple) is patterned on a high-refractive-index substrate (pink). In this waveguide structure, a TM bound mode will have effective coupling with the TE continuous modes. The loss of the TM bound mode to the TE continuum occurs at the two waveguide edges as illustrated in Fig. R1. The loss at each edge originates from the coupling of the TM bound mode with the left-going (Channels 1 and 3) and right-going (Channels 2 and 4) TE continuous modes. If the losses via Channels 1 (2) and 3 (4) interfere destructively and cancel each other out, then the total loss of the TM bound mode to the TE continuum can be reduced to zero, leading to a lossless TM bound mode which is the desired BIC. The inference of losses via Channels 1 (2) and 3 (4) depends on the phase difference caused by the finite width of the waveguide, so the BIC can be obtained just by optimizing the waveguide width w . It should be noted that the above analysis is suitable for the TM bound modes of any order. The only difference among the TM BIC modes of different orders is their coupling with the TE continuous modes, so they require different waveguide widths to achieve the zero propagation loss.

Fig. R1. Radiation channels of the TM bound mode to the TE continuum.

Revision details

- **Supplementary Information, Page 2, Paragraph 2, Line 4–14**

The loss of the TM bound mode to the TE continuum occurs at the two waveguide edges as illustrated in Fig. S2. The loss at each edge originates from the coupling of the TM bound mode with the left-going (Channels 1 and 3) and right-going (Channels 2 and 4) TE

continuous modes. If the losses via Channels 1 (2) and 3 (4) interfere destructively and cancel each other out, then the total loss of the TM bound mode to the TE continuum can be reduced to zero, leading to a lossless TM bound mode which is the desired BIC. The inference of losses via Channels 1 (2) and 3 (4) depends on the phase difference caused by the finite width of the waveguide, so the BIC can be obtained just by optimizing the waveguide width w . It should be noted that the above analysis applies generally to the TM bound modes of any order. The only difference among the TM BIC modes of different orders lies in their coupling with the TE continuous modes, so they require different waveguide widths to achieve the zero propagation loss.

- **Supplementary Information, Figure S2 in the Supplementary Information**

Comment 2: *In Figure 1, which is supposed to give the reader an overview of the message of the paper, there is no information about BIC.*

Response: Thanks for the reviewer's valuable comment. Actually, in Figs. 1a–1d, the conveyed message is that the TM modes of different orders (Figs. 1b and 1c) cannot propagate with negligible loss in conventional wisdom because of their inevitable coupling with the TE continuous modes (Fig. 1d). The BIC mechanism promises that zero propagation loss can be obtained for the TM modes of any order just by engineering the waveguide width w . Therefore, Fig. 1e shows the propagation loss as a function of the waveguide width for the TM modes of 4 different orders. If the propagation loss of a TM mode is zero for a certain waveguide width, then that TM mode is the exact BIC as we desire. If the propagation loss of a TM mode is nonzero but is small enough for practical applications, then that TM mode is the quasi-BIC. Therefore, Fig. 1e has clearly shown that the BIC can be obtained for the TM modes of each order in the waveguide structure shown in Fig. 1a by engineering the waveguide width w . In conclusion, Fig. 1 has delivered the message of why we need to find and how to obtain the BIC in the TM modes of different orders in the waveguide.

Revision details

- **Main manuscript, Results, Paragraph 1, Last line 1–4**

This means that the BICs can be obtained for the TM bound modes in each order just by engineering the waveguide width. In addition, the tolerance of waveguide width for the BICs in each order to maintain negligible propagation loss is as large as several hundreds of nanometers, which can well be accommodated in device fabrication.

- **Main manuscript, Figure 1e**

Comment 3: *Authors refer to increasing the capacity for increasing demand for huge data transmission (and name the essential techniques) to raise the point that data multiplexing is essential. Then it is noted that these multiplexing techniques have been mostly demonstrated in SOI platform - which is not true (unless they mean for on-chip platforms, which is again not a strong argument). I do not think that there is yet an on-chip solution in commercial use in, for example, any data center. At the end of the paragraph, they motivate the use of LNOI platform in this work by stating its outstanding properties; however, they do not take advantage of this platform at all. The application of LNOI for them is just as a substrate with a larger refractive index than the polymer. Also, (1) proved that LNOI platform has small*

optical loss down to 2.7 dB/m which is better than the numbers derived from simulations of this work; therefore, the etching issues is also not relevant.

[1] M. Zhang, C. Wang, R. Cheng, A. Shams-Ansari, and M. Lončar, "Monolithic ultra-high- Q lithium niobate microring resonator," *Optica*, vol. 4, no. 12, pp. 1536–1537, 2017.

Response: Thanks for the reviewer's valuable comment.

- First, we are sorry for the misleading statement in our original manuscript. It should be pointed out that the multiplexing techniques mentioned in our original manuscript refer to on-chip platforms. We have made revisions in the revised manuscript to make this point clear.
- Second, we agree with the reviewer that there has on-chip solution in commercial use. However, the existing solution based on silicon photonics is not the best. For example, silicon modulators requiring p-n doping have not achieve a modulation speed beyond 100 GHz as in LiNbO₃. Also, it has many other disadvantages as we discussed in the introduction of our manuscript. Therefore, exploring on-chip solutions based on other materials like LiNbO₃ may push this technology to a higher level.
- Third, in order to prove that our method can utilize the advantages of LiNbO₃, we have conducted additional experiments to achieve modulation and mode (de)multiplexing on a single chip. Figures 6a and 6b in the revised manuscript show the fabricated device and experimental setup. Figure 6c in the revised manuscript plots the measured modulation signal of each channel. It should be noted that the modulation speed here is limited by the on-chip electro-optic modulator instead of the mode (de)multiplexer. The data transmission capacity of our mode (de)multiplexer has been proven independently as shown in Fig. 6e in the revised manuscript. It should be noted that this work focuses mainly on high-order BICs rather than demonstrating high-speed modulation. Achieving both high-speed modulation and mode (de)multiplexing on a single chip would be the next step of our research effort.
- Last, it should be noted that the optical losses of the high-order modes are mainly from the mode coupling rather than the propagation. Actually, the propagation loss does not play a major role in our device performance. Additionally, the propagation loss of high-order modes has not been discussed in paper (1) yet. More importantly, it would be unfair to compare the performance of our newly developed platform with a mature platform that has been developed tens of years. The striking advantage of our newly developed platform is that it provides a practical and reliable method to explore functional single-crystal materials (not specific to LiNbO₃) in integrated photonics. For example, barium titanate (BTO) has the Pockels coefficient ~ 30 times larger than LiNbO₃ [Nat. Materials **18**, 42–47 (2019)], and thus is an ideal material for on-chip electro-optic modulation in optical communication. It is expected to have extremely small power consumption and ultrahigh electrical-to-optical or optical-to-electrical conversion efficiencies. However, to date, there is still no effective etching method for patterning microstructures in BTO. Our newly developed platform based on the BIC principle can overcome the problem by using a fabrication-friendly material (e.g., Si₃N₄) on the BTO substrate.

Revision details

- **Main manuscript, Results, Last paragraph, Line 1–13**

Finally, we integrated electro-optic modulators with the mode (de)multiplexer on the same chip as shown in Fig. 6a to demonstrate the capability of on-chip electro-optic modulation and mode (de)multiplexing. Light sent into the four input channels was first modulated by a microcavity electro-optic modulator in each channel, then passed through the mode (de)multiplexer before being directed to the corresponding output channels. Figure 6b shows the experimental setup, where the light from a semiconductor laser with wavelength of ~ 1.55 μm was amplified by an erbium-doped fiber amplifier (EDFA) and then sent through a fiber polarization controller before being coupled into the device under test. An electrical driving signal from a signal generator was amplified and then applied to the device under test via a microwave (MW) probe. The light transmitted through the device was amplified by a second EDFA and then collected by a high-speed photodetector which converted the detected signal into the electrical domain for monitoring on an oscilloscope. Figure 6c shows the measured signals for the input and output pairs of TM_0 - TM_0 , TM_1 - TM_1 , TM_2 - TM_2 , and TM_3 - TM_3 , demonstrating that our fabricated device could achieve both electro-optic modulation and mode (de)multiplexing on a single chip.

- **Main manuscript, Figures 6a–6c in the main manuscript.**

Comment 4: *There exist quite a few papers that implemented the same idea as this paper, such as (2; 3) that they use Silicon Nitride (SiN) on top of LNOI, and they show fast modulations. At the same time, they are other articles such as (4; 5; 6; 7) that use the similar idea of mode coupling (by changing the width of the through-port, gap, and the waveguide (with or without rings). This paper has well combined these two techniques, but the performance is not better than other platforms.*

- [2] A. N. R. Ahmed, S. Shi, M. Zablocki, P. Yao, and D. W. Prather, "Tunable hybrid silicon nitride and thin-film lithium niobate electro-optic microresonator," *Optics letters*, vol. 44, no. 3, pp. 618–621, 2019.
- [3] L. Chang, M. H. Pfeiffer, N. Volet, M. Zervas, J. D. Peters, C. L. Manganelli, E. J. Stanton, Y. Li, T. J. Kippenberg, and J. E. Bowers, "Heterogeneous integration of lithium niobate and silicon nitride waveguides for wafer-scale photonic integrated circuits on silicon," *Optics letters*, vol. 42, no. 4, pp. 803–806, 2017.
- [4] L.-W. Luo, N. Ophir, C. P. Chen, L. H. Gabrielli, C. B. Poitras, K. Bergmen, and M. Lipson, "WDM-compatible mode-division multiplexing on a silicon chip," *Nature communications*, vol. 5, p. 3069, 2014.
- [5] B. Stern, X. Zhu, C. P. Chen, L. D. Tzuang, J. Cardenas, K. Bergman, and M. Lipson, "On-chip mode-division multiplexing switch," *Optica*, vol. 2, no. 6, pp. 530–535, 2015.
- [6] J. Wang, P. Chen, S. Chen, Y. Shi, and D. Dai, "Improved 8-channel silicon mode demultiplexer with grating polarizers," *Optics express*, vol. 22, no. 11, pp. 12799–12807, 2014.
- [7] Y.-D. Yang, Y. Li, Y.-Z. Huang, and A. W. Poon, "Silicon nitride three-mode division multiplexing and wavelength-division multiplexing using asymmetrical directional

couplers and microring resonators," Optics express, vol. 22, no. 18, pp. 22172–22183, 2014.

Response: Thanks for the reviewer’s comment. However, we respectfully disagree that the papers (2;3) implemented the same idea as our paper. Below please find our argument:

- First, the waveguide material in papers (2;3) is silicon nitride ($n \approx 2.0$), which has a similar refractive index as LiNbO_3 ($n_e \approx 2.13$). This is just a special case applicable to the LiNbO_3 platform, and cannot be applied generically to other material platforms because it is not based on the BIC mechanism. In our work, we adopt a low-refractive-index material ($n = 1.54$) on a high-refractive-index ($n = 2.13$) platform, with a large refractive index contrast between the waveguide and the substrate. Based on the BIC mechanism, the method in our work can be applied to any other material platforms, such as silica on BTO or YIG. It should be noted that BTO, YIG, and many other single-crystal materials do not have any effective method for high-quality etching yet.
- Second, without the BIC mechanism, a low-refractive-index waveguide on a high-refractive-index substrate would have very limited applications. Actually, the modes utilized in papers (2;3) are both the TE modes, because only the TE modes can be utilized in those cases. The TM mode would suffer serious propagation loss unless the waveguide structure is engineered under the BIC mechanism. Based on the parameters provided in paper (2) (200-nm Si_3N_4 on 300-nm LiNbO_3), the propagation loss of the TM mode in a straight waveguide as a function of the waveguide width is plotted in Fig. R2a, where the insets show the modal profiles of the TM modes for waveguide with different waveguide widths. It is clear that the TM mode could actually be utilized if the waveguide width had been optimized based on the BIC principle. It should be noted that a disadvantage of using the TE mode is the weak effective refractive index contrast as shown in Fig. S1 in the revised Supplementary Information. The TE modes can be effectively guided only in a straight waveguide, because a bent waveguide would require a radius of hundreds of micrometers to suppress the bending loss as evidenced in paper (2). A bent waveguide with such a large radius is similar to a straight waveguide. For a bent waveguide with optimized waveguide width of 1.75 μm , Fig. R2b plots the propagation loss of the TM mode as a function of the radius, which clearly shows that the bent waveguide can actually achieve low propagation loss with much smaller bend radius compared with that in paper (2). Therefore, our waveguide structure under the BIC mechanism can maintain low propagation loss in both the straight and bent forms with a compact size, which has clear advantages in enhancing integration density of photonic integrated circuits.

In summary, our work has explored and harnessed the exciting BIC physical mechanism of a low-refractive-index waveguide on a high-refractive-index substrate, which makes it fundamentally different from those in papers (2;3) and produces significantly broader application areas.

On the other hand, we believe that it is unfair to compare the performance of mode (de)multiplexing of our paper with those on the silicon platform. It should be noted that the good performance in (4;5;6;7) is attributed to the large refractive index ($n \approx 3.46$) of silicon. To our knowledge, high-order mode manipulation has not yet been demonstrated in materials with a much lower refractive index ($n \approx 2.1$) such as AlN , Si_3N_4 , and LiNbO_3 . Therefore, our paper is actually the first demonstration of mode (de)multiplexing on a material platform with

a much lower refractive index. Additionally, we believe there is still room for further improvement of the performance of our device if asymmetric couplers as in [Opt. Express **21**, 8, 10376–10382 (2013)] are adopted.

Fig. R2. (a) Propagation loss of the TM mode in a straight waveguide as a function of the waveguide width. (b) Loss of a 90° waveguide bend with optimized waveguide width of 1.75 μm as a function of the bend radius.

Comment 5: Authors neglect to provide details on the proposed design (which all can be in supplementary). For example, they forget to mention what gaps they used for the design (g_1, g_2, g_3 , and g_4). There are also no details on the parameters of the grating couplers and their insertion loss and performance. On the simulation side, I could not find any details on how the propagation loss in figure.1 was calculated. There are also some details of fabrication and measurement missing (E-beam resist, measurement details).

Response: Thanks for the reviewer’s valuable comments. We have provided the missing information in the revised manuscript.

Revision details

- **Main manuscript, Page 4, Last paragraph, Line 1–2**

To obtain the optimal coupling length of the multimode directional couplers, we set g_1 , g_2 , and g_3 to be 550, 550, and 450 nm, respectively.

- **Main manuscript, Methods**

Simulation. The effective refractive index and propagation loss were simulated for each mode in the BIC waveguide by using a finite-element method in COMSOL. Eigenmode analysis was adopted in simulating the real and imaginary part of the effective refractive index for each mode. The propagation loss was obtained from the complex effective refractive index.

Device fabrication. We fabricated all the devices on a LiNbO₃-on-insulator wafer (NANOLN), with a 400-nm-thick LiNbO₃ layer on 2-μm-thick silicon oxide. For the passive devices without the function of electro-optic modulation, the fabrication process consists of only one step of electron-beam lithography, which defined the patterns in a 500-nm-thick polymer (electron-beam resist ZEP520A). For the active devices with the function of electro-optic modulation, we first fabricated the electrodes with a lift-off process involving electron-

beam lithography and gold deposition, where the thickness of the gold electrodes was 80 nm. Then, we spin-coated a 500-nm-thick layer of polymer (ZEP520A) and performed a second step of electron-beam lithography to pattern the photonic waveguides and grating couplers in the polymer.

- **Supplementary Information, Last paragraph**

Figure S5a is an optical microscope image of a fabricated device with two grating couplers connected by a straight waveguide. Figure S5b is a scanning electron microscope image zoomed in at the grating coupler, which has a grating period of 1.15 μm and duty cycle of 0.5. Figure S5c plots the measured optical transmission spectrum of a single grating coupler, which shows that the minimal insertion loss of a single grating coupler is less than 12 dB.

- **Supplementary Information, Figure S5**

Comment 6: *In the end, for the eye diagram measurement, it is necessary to measure the cross-talk rigorously, writing a number is not sufficient.*

Response: Thanks for the reviewer’s valuable comment. First, we agree that rigorous measurement of crosstalk is necessary. However, the main focus of our paper is on the high-order BIC modes, which includes the principle for obtaining the high-order BIC modes, experimental demonstration of the high-order BIC modes, and proof-of-concept of mode (de)multiplexing with these high-order BIC modes. We believe that the contents in our paper have already provided a self-consistent, rigorous, and thorough investigation of high-order BICs. Demonstration of high-speed on-chip communication and rigorous measurement of the modulation performance require a sophisticated measurement system. They deviate from the main focus of this work and are beyond the scope of this paper. We have plans to carry out these tasks in the next phase and publish the relevant results elsewhere.

Here, we analyzed the influence of crosstalk to the eye diagrams to prove our device can work in real applications. In our experiment, the crosstalk for each channel is introduced by the other three channels. For example, the modulated signal in the TM_2 – TM_2 channel will be influenced by the input signals in the TM_0 , TM_1 , and TM_3 channels, because the signals from the TM_0 , TM_1 , and TM_3 channels can also be coupled slightly into the TM_2 channel. These undesired signals will affect the purity of signal in the TM_2 – TM_2 channel. The influence of such crosstalk on eye diagram was numerically calculated based on the experimentally measured eye diagrams in Fig. 6e of the revised manuscript and the crosstalk in Figs. 5b–5e of the revised manuscript. Figures S4a– S4e in the revised Supplementary Information plot the simulated eye diagrams in the TM_0 , TM_1 , TM_2 , and TM_3 output channels with no crosstalk, with crosstalk as the measured results in Figs. 5b–5e of the revised manuscript, –10-dB crosstalk between any two channels, –7-dB crosstalk between any two channels, and –5-dB crosstalk between any two channels, respectively. A comparison between Figs. S4a and S4b in the revised Supplementary Information concludes that the experimental crosstalk has negligible influence on the measured eye diagrams. Figures S4c–S4e in the revised Supplementary Information show that the eye diagrams start to be affected when the crosstalk is –7 dB between any two channels and have almost closed eyes when the crosstalk increases

to -5 dB between any two channels. These results indicate that our fabricated mode (de)multiplexer can support 40-Gbps/channel data transmission in real applications.

Revision details

- **Supplementary Information, Page 3, Last paragraph**

Here, we analyzed the influence of crosstalk on the eye diagrams to prove our device can work in real applications. In our experiment, the crosstalk for each channel was introduced by the other three channels. For example, the modulated signal in the TM_2 - TM_2 channel could be influenced by the signals input into the TM_0 , TM_1 , and TM_3 channels, because the signals from the TM_0 , TM_1 , and TM_3 channels could also be coupled slightly into the TM_2 channel. These undesired signals would affect the purity of signal in the TM_2 - TM_2 channel. The influence of such crosstalk on eye diagrams was numerically calculated based on the experimentally measured eye diagrams in Fig. 6e of the main manuscript and the crosstalk in Figs. 5b–5e of the main manuscript. Figures S4a– S4e plot the simulated eye diagrams in the TM_0 , TM_1 , TM_2 , and TM_3 output channels with no crosstalk, with crosstalk as the measured results in Figs. 5b–5e of the main manuscript, with crosstalk of -10 , -7 , and -5 dB between any two channels, respectively. A comparison between Figs. S4a and S4b concludes that the experimental crosstalk has negligible influence on the measured eye diagrams. Figures S4c– S4e show that the eye diagrams start to be affected when the crosstalk is -7 dB between any two channels and have almost closed eyes when the crosstalk increases to -5 dB between any two channels. These results indicate that our fabricated mode (de)multiplexer can support 40-Gbps/channel data transmission in real applications.

- **Supplementary Information, Figure S4**

Response to Reviewer 3 ---- NCOMMS-19-27126

The manuscript under review is dedicated to the design and experimental demonstration of a 4-channel mode division multiplexer exploiting an unconventional (BIC-based) guiding mechanism. The authors design a multimode polymer waveguide on the lithium niobate platform, in which all the four supported modes are near the BIC condition at different "BIC orders" and thus have low propagation loss and high propagation length. On the basis of this waveguide, the authors design and experimentally demonstrate a 4-channel multiplexer/demultiplexer chip. The fabricated structure has low insertion loss and low channel crosstalk within a relatively broad wavelength range of 70 nm. The fabricated structure, being tested with 40 Gbps/channel data transmission, demonstrates an eye diagram having reasonable quality.

I believe that this paper is, in principle, suitable for publication in Nature Communications since it presents a new important practical application of the BIC effect and, for the first time, demonstrates the possibility of simultaneously using several BICs (or, strictly speaking, several near-BIC modes) of different orders. In addition, the proposed approach allows one to avoid the patterning of the LiNbO₃ layer, which is quite challenging.

However, in my opinion, the manuscript requires a minor revision before it can be accepted for publication. My comments are given below.

Response: We thank the reviewer for reading and carefully reviewing our work. We are delighted to see that the reviewer understands and appreciates our work, as shown in his/her comments “suitable for publication in Nature Communications”, “it presents a new important practical application of the BIC effect”, and “for the first time, demonstrates the possibility of simultaneously using several BICs”.

In what follows, we will provide detailed responses to the individual comments.

Comment 1: *Bound states in the continuum in ridge structures very similar to the one used as a building block for the proposed (de)multiplexer have been comprehensively studied by several groups. Therefore, in order to place this work in a proper context, I would suggest citing and briefly discussing in the introduction at least the following papers in addition to the cited paper by Zou et al. (Ref. [5] in the manuscript):*

- A. P. Hope, T. G. Nguyen, A. Mitchell, and W. Bogaerts. Quantitative analysis of TM lateral leakage in foundry fabricated silicon rib waveguides. IEEE Photonics Technol. Lett., 28(4):493–496, 2016.

- E. A. Bezus, D. A. Bykov, and L. L. Doskolovich. Bound states in the continuum and high-Q resonances supported by a dielectric ridge on a slab waveguide. Photonics Research, 6(11):1084–1093, 2018.

- T. G. Nguyen, G. Ren, S. Schoenhardt, M. Knoerzer, A. Boes, and A. Mitchell. Ridge resonance in silicon photonics harnessing bound states in the continuum. Laser & Photonics Reviews, (in print):1900035, 2019.

Response: Thanks for the reviewer’s valuable suggestions. We have added more citations and discussion about BICs in integrated photonics in the introduction in order to let readers have an overview of BICs.

Revision details (Page 2, Paragraph 1, Line 6–8)

...and integrated photonics^[13,14,18-21]. For integrated photonics, resonances can be found in a single rib waveguide without any cavity structure due to the BIC mechanism^[19-21]. In addition,...

Comment 2: *Probably, it is worth showing the numerical and experimental results for the transmission of the designed multimode interference couplers (Figs. 3 and 4) on the same plots. It would give the reader a better idea of how close the obtained experimental results are to the results predicted numerically.*

Response: Thanks for the reviewer's comment. We considered grouping Figs. 3 (simulated results) and 4 (measured results) into one during manuscript preparation. However, we found such a combined figure would be too large without reducing the font sizes of the labels inside. Therefore, we have to leave the figures as they are for better presentation quality.

Comment 3: The authors should comment on the outlook of this work. Can the number of (de)multiplexed channels be increased, or it becomes impossible to maintain the near-BIC condition for a larger number of modes?

Response: Thanks for the reviewer's comment. We believe that the number of (de)multiplexed channels can be increased based on our design rules.

- First, the BIC mechanism is generic for modes of all orders as analyzed in Supplementary Information (Page 2, Paragraph 2), so it is possible to find the BICs for modes of each order.
- Second, once we fix the waveguide width for the fundamental mode, we can then find the waveguide width for all the higher-order modes to achieve the phase-matching condition, because the effective refractive indices of modes in all orders depend on the waveguide widths.
- Third, higher-order modes require a wider waveguide width to achieve the phase-matching condition. The propagation loss of the BIC waveguide for modes in each order decreases significantly as the waveguide width increases. In Fig. 2b in the revised manuscript, we can find that all the high-order modes require a quite large waveguide width to achieve the phase-matching condition. All these high-order modes can maintain relatively low propagation loss with these waveguide widths.

Therefore, the number of (de)multiplexed channels can be increased. We have added the outlook in the revised manuscript which is highlighted in red.

Revision details (Conclusion, Last line 4–6)

The number of (de)multiplexing channels can be further increased because the BICs for higher-order modes and phase-matching condition can be satisfied simultaneously based on the design method employed in this work.

Comment 4: *In the conclusion, the authors state that the demonstrated structure can be used in a "hybrid MDM-PDM-WDM optical link", but do not demonstrate the performance of their structure in the case of TE-polarization. It is clear that quasi-TE modes of the polymer*

photonic rib waveguide will be confined by the index guiding mechanism, however, a short discussion (probably, in the Supporting Information) would benefit the paper.

Response: Thanks for the reviewer's very careful comment. We made a mistake on this statement. Actually, the TE mode cannot be guided by such a waveguide because of the low effective-refractive-index contrast as shown in Fig. S1a. We have modified our statement in the revised manuscript as "hybrid MDM-WDM optical link".

Revision details (Conclusion, Last line 2)

...hybrid MDM-WDM optical link...

Reviewers' Comments:

Reviewer #1:

Remarks to the Author:

I am satisfied by revisions which meet comments 1 and 2.

However I am confused by response to the comment 3.

Once more history of BICs.

For a long time the localized solutions with discrete energies embedded into the continuum discovered by von Neumann and Wigner were considered as mathematical curiosity because hardly such potentials can be realized in reality. The decisive breakthrough came with paper by Friedrich and Wintgen in 1985 (PRA 32, 3231) which formulated the concept of the BIC as the result of complete destructive interference of two resonances undergoing an avoided crossing.

This concept was applied to microwave resonator in 2006 by Sadreev et al , PRB 73, 235342 and by Bulgakov et al in 2008 to 2D photonic crystal (PRB 78, 075105).

The last paper as well as the paper by Marinica et al (PRL 100, 183902 [2]) were pioneering about BICs in photonics which

stimulated further a mass of theoretical and experimental papers.

The reference [3] by S. Longhi is not relevant. Better to cite the paper by Longhi et al (PRL,111, 220403 (2013)) with experimental realization of BICs in 1d array of waveguides.

Reviewer #2:

None

Reviewer #3:

Remarks to the Author:

The authors have adequately addressed the comments presented in my initial review, so, in my opinion, the manuscript can now be accepted for publication.

Response to Reviewer 1 ---- NCOMMS-19-27126A

I am satisfied by revisions which meet comments 1 and 2. However I am confused by response to the comment 3. Once more history of BICs. For a long time the localized solutions with discrete energies embedded into the continuum discovered by von Neumann and Wigner were considered as mathematical curiosity because hardly such potentials can be realized in reality. The decisive breakthrough came with paper by Friedrich and Wintgen in 1985 (PRA 32, 3231) which formulated the concept of the BIC as the result of complete destructive interference of two resonances undergoing an avoided crossing. This concept was applied to microwave resonator in 2006 by Sadreev et al , PRB 73, 235342 and by Bulgakov et al in 2008 to 2D photonic crystal (PRB 78, 075105). The last paper as well as the paper by Marinica et al (PRL 100, 183902 [2]) were pioneering about BICs in photonics which stimulated further a mass of theoretical and experimental papers. The reference [3] by S. Longhi is not relevant. Better to cite the paper by Longhi et al (PRL,111, 220403 (2013)) with experimental realization of BICs in 1d array of waveguides.

Response: We appreciate that the reviewer provided a detailed history of development of BIC. As for the references in the last version of our manuscript, Refs. [1-8] were some historic works of BICs, and Refs. [9-14] were some experimental breakthroughs of BICs in the photonic domain due to advancement of nanofabrication technology. To be more accurate, we have added new references about the BIC history in the revised manuscript according to the reviewer's advice.

Revisions:

- **Main manuscript, Page 2, Paragraph 1, Line 2**
... can coexist with continuous waves without any radiation loss^[1-9].
- **Main manuscript, Page 2, Paragraph 1, Line 5**
... has triggered fast development of BICs in photonics^[10-16],